# Construction of a Prognostic and Early Diagnosis Model for LUAD Based on Necroptosis Gene Signature and Exploration of Immunotherapy Potential

**DOI:** 10.3390/cancers14205153

**Published:** 2022-10-20

**Authors:** Baizhuo Zhang, Yudong Wang, Xiaozhu Zhou, Zhen Zhang, Haoyu Ju, Xiaoqi Diao, Jiaoqi Wu, Jing Zhang

**Affiliations:** 1Department of Pharmacology, College of Pharmacy, China Medical University, Shenyang 110122, China; 2Thoracic Surgery Department, Shengjing Hospital of China Medical University, Shenyang 110004, China

**Keywords:** necroptosis, lung adenocarcinoma, biomarker, tumor immunotherapy, immune microenvironment, drug sensitivity, immune checkpoint, prognostic and diagnostic

## Abstract

**Simple Summary:**

Necroptosis plays an important role in the progression and metastasis of lung adenocarcinoma (LUAD) and regulates the inflammatory response and tumor microenvironment. First of all, through NRGs, we determined the LUAD early diagnosis model, which is composed of four necroptosis-related genes (NRGs) (AUC = 0.994), and the LUAD prognosis evaluation model, composed of nine NRGs (AUC = 0.826). Secondly, the LUAD prognosis model was found to be closely related to immune checkpoint inhibitor (ICI) treatment and chemosensitivity. ICI treatment is more suitable for low-risk patients, while chemotherapy is more effective for high-risk patients. Finally, we identified the core gene PANX1 for the first time, which is important for prognosis evaluation and early diagnosis, and analyzed its role in LUAD immunotherapy. This study provides a new target for the immunotherapy of LUAD and a new theoretical basis for future individualized treatment in the clinic.

**Abstract:**

Necroptosis is a type of programmed necrosis that is different from apoptosis and necrosis. Lung cancer has the highest incidence and mortality worldwide, and lung adenocarcinoma is the most common subtype of lung cancer. However, the role of necroptosis in the occurrence and development of LUAD remains largely unexplored. In this paper, four NRGs and nine NRGs determined by big data analysis were used to effectively predict the risk of early LUAD (AUC = 0.994) and evaluate the prognostic effect on LUAD patients (AUC = 0.826). Meanwhile, ESTIMATE, single-sample gene set enrichment analysis (ssGSEA), genomic variation analysis (GSVA), gene set enrichment analysis (GSEA), and immune checkpoint analysis were used to explore the enrichment characteristics and immune research related to the prognostic model. In deep data mining, we were surprised to find that prognostic models also regulate the immune microenvironment, cell cycle, and DNA damage repair mechanisms. Thus, we demonstrated a significant correlation between model evaluation results, ICI treatment, and chemotherapeutic drug sensitivity. The low-risk population has a stronger tumor immune response, and the potential for ICI treatment is greater. People at high risk respond less to immunotherapy but respond well to chemotherapy drugs. In addition, PANX1, a core gene with important value in immune regulation, prognosis assessment, and early diagnosis, has been identified for the first time, which provides a new target for the immunotherapy of LUAD as well as a new theoretical basis for the basic research, clinical diagnosis, and individualized treatment of LUAD.

## 1. Introduction

Lung cancer is the most common malignancy and is the leading cause of cancer deaths worldwide [1,2]. Non-small cell lung cancer (NSCLC) accounts for nearly 80% of lung cancers, of which approximately 50% are LUAD. In recent years, with advances in diagnostic, surgical, radiotherapy, and molecular-targeted therapy techniques, the clinical outcomes of patients with LUAD have improved significantly. However, cancer statistics show that the 5-year survival rate of patients with LUAD remains low [3,4,5,6]. Therefore, there is an urgent need to find novel perspectives to screen new targets and construct predictive models that can be used for LUAD diagnosis, prognosis, immunotherapy, and chemotherapeutic drug screening.

Since most tumors are inborn and resistant to apoptosis, the process of inducing cell death pathways, such as necroptosis, has gradually been recognized as a potential therapeutic approach [7]. Necroptosis is a novel form of programmed cell death distinct from apoptosis that enhances CD8+ T cell-mediated anti-tumor immunity via a mechanism involving RIPK3 and RIPK1 activation in the tumor microenvironment (TME) [8]. Necroptosis has been proposed as a potential target for immunotherapy in LUAD. The role of necroptosis in various diseases has been made increasingly apparent by recent studies. For example, knocking down RIPK1, RIPK3, and MLKL in colorectal and esophageal cancer cells inhibits tumor growth by reducing NF-κB activity [9]. Research has demonstrated that necroptosis is able to overcome resistance to cancer drugs mediated by P-glycoprotein, Bcl-2, and Bcl-xL in cancer cell lines [10]. When cysteine is inhibited, necroptosis can resist pathogens and cancer cells escaping from apoptosis, which is also one of the key mechanisms of several anti-tumor drugs such as 5-FU, etoposide, and cisplatin [11]. In addition, necroptosis can also change the tumor immune microenvironment by regulating immune checkpoints [12]. All these results suggest that necroptosis may be an effective therapeutic target for cancer. Some NRGs have recently been identified as prognostic markers for cancer and other diseases [13]. Despite this, the exact role of NRGs in LUAD is unclear. Therefore, understanding the impact of NRGs on the occurrence and development of LUAD can provide potential biomarkers and therapeutic targets as well as guide the immunotherapeutic strategy for LUAD.

In this study, we aim to develop a prediction model related to necroptosis genes, which is of guiding significance in the clinical diagnosis, prognosis evaluation, immunotherapy, and drug screening of LUAD. First, we screened NRGs and calculated prognostic risk scores using univariate Cox, Lasso, and multivariate Cox regression. Secondly, we combined multiple clinical features to construct a nomogram and verified them through calibration evaluation to quantitatively predict the prognosis of LUAD patients. In addition, we explored the NRGs closely related to the early development of LUAD in the prognosis model and constructed a new diagnosis model, which generated new directions for the clinical research of dual biomarkers with diagnostic and prognostic potential in LUAD. Finally, we found that PANX1 may become a new target of LUAD immunotherapy through multiple big data analyses and has good predictive ability for future clinical diagnosis, prognosis, and immunotherapy.

## 2. Materials and Methods

### 2.1. Summary of Necrotizing Apoptosis Genes

The overall design and flowchart for this study are shown in Figure 1. Based on previous literature reports, 150 NRGs were summarized. A total of 159 and 614 NRGs were subsequently retrieved from the Kyoto Encyclopedia of Genes and Genomes (KEGG, https://www.kegg.jp/) (accessed on 1 June 2022) and the comprehensive database of human genes (GeneCards, https://www.genecards.org/) (accessed on 1 June 2022). These three methods were used to aggregate 767 non-duplicate NRGs for further study and analysis (see Appendix A).

### 2.2. Open Data Sets and Pre-Processing

The genome-related database UCSC Xena (http://xena.ucsc.edu/) (accessed on 4 June 2022) includes the functionality of many oncology research databases and provides visual analysis for public data centers. To reduce errors in subsequent differential analyses due to the small number of control samples in The Cancer Genome Atlas (TCGA), we downloaded gene expression data and relevant clinical information from TCGA and Genotype-Tissue Expression (GTEx) through UCSC Xena. The TCGA LUAD dataset contained 526 LUAD and 59 non-cancer control samples, whereas the GTEx dataset contained 288 healthy human samples. Consequently, this study included 526 cases of LUAD and 327 non-cancer cases. In recent years, the resulting data from worldwide research studies have shown that the incidence of lung cancer is relatively low up to the age of 40 years and increases sharply thereafter, peaking in both male and female populations in the 80 to 84 age group, and that the overall lung cancer incidence rate is higher in men than in women. Therefore, in addition to survival time and survival status, factors such as age and gender were included in the clinical information comprising the statistics in our study. Detailed clinical information is shown in Table 1, including age, sex, pathological stage, TNM stage, and survival status. Gene expression profiles and the corresponding clinical information for two microarrays, GSE75037 and GSE19188, were downloaded from the Gene Expression Omnibus database (GEO, www.ncbi.nlm.nih.gov/geo) (accessed on 4 June 2022). The two microarrays were GPL6884 and GPL570. GSE75037 contained 83 LUAD tissues and 83 matched non-malignant lung tissues. GSE19188 contained 45 LUAD tissues and 65 adjacent standard lung tissues. Finally, a dataset of mutations and copy number variants in LUAD was downloaded from UCSC Xena and included 531 samples. (Since the grade data in TCGA-LUAD, GEO-GSE75037, and GEO-GSE19188 were missing, we removed the grade category from the clinical information in Table 1.)

### 2.3. Identification of Differentially Expressed Genes in NRGs and the Related Enrichment Analysis

We extracted expression data for 767 NRGs from TCGA and evaluated 734 differentially expressed necroptosis genes (DENRGs) based on FDR < 0.05 and |log2FC| ≥ 1. The R package “pheatmap” was used to generate heat maps representing differential expressions. DENRG functions were uncovered using the R package “clusterProfiler” for gene ontology (GO) and for the Kyoto Encyclopedia of Genes and Genomes (KEGG) pathway functional enrichment analysis in order to assign the biological processes, cellular components, molecular functions, and pathway clustering of identified marker genes.

### 2.4. Construction and Validation of Prognostic Models

First, the entire TCGA dataset was randomly divided into a training set and a test set at a 7:3 ratio through the R package “caret”. Next, a Lasso Cox regression analysis was performed using the R packages “glmnet” and “survival”. Lastly, we conducted a multivariate Cox regression analysis of DENRGs derived from lasso regression screening to determine the following risk score formula for LUAD:Risk Score = ∑ n t = 1 Coefi × Xi.

Coefi is the correlation coefficient for each DENRG, and X is the gene expression level. The model’s predictive power was analyzed using the Kaplan-Meier (K-M) method, receiver operating characteristic (ROC), and principal component analysis (PCA). The R package “survival” was used to analyze the difference in overall survival time (OS) between the low- and high-risk groups, and the K-M survival curves were plotted based on the Wilcoxon test. To assess the sensitivity and specificity of the model, the R packages “timeROC”, “ROCR”, and “survival” were used to plot the ROC curves at 1, 3, and 5 years. PCA was performed utilizing the R packages “limma” and “scatterplot3d” to show the distribution patterns of high-risk and low-risk populations.

### 2.5. Relationship between Risk Score and Independent Prognosis

Using the R package “survival”, we conducted univariate and multivariate Cox regression analyses to determine whether risk scores and multiple clinical characteristics could be considered independently as prognostic factors, and the R packages “survival”, “pec”, and “rms” were used to construct a concordance index (C-index) to evaluate the predictive accuracy of the prognostic models. A chi-square test was used to compare the differences in clinicopathological characteristics between different risk groups, and then a heat map of the correlation between prognostic risk scores and clinical features was created using the R package “pheatmap”.

### 2.6. Nomogram Construction and Verification

Based on the results of the multifactorial Cox proportional risk regression analysis, a nomogram was constructed using the R package “rms” to predict patient survival at 1, 3, and 5 years, followed by the annotation of patient risk score information for clinical use. To assess the accuracy of the nomogram, calibration curves, multi-indicator ROC curves, and Decision Curve Analysis (DCA) were plotted via the R packages “survivor”, “survminer”, “timeROC”, and “ggDCA”.

### 2.7. Relationships between Risk Score, Gene Set Enrichment Analysis (GSEA), and Mutations

In the LUAD prognostic model, we explored the signaling pathways associated with different risk groups using the GSEA software (GSEA 4.2.3). The multipleGSEA plot shows the potential activation pathways in both risk groups in the part of the pathway predominantly enriched by GO and KEGG. To demonstrate the importance of genomic and pathway correlations, normalized enrichment scores, nominal *p* values, and false discovery rate q values were calculated. The “maftool” R package was used to analyze the mutation patterns and the number of mutations in the two risk groups.

### 2.8. Risk Score and Immunization Correlation Analysis

The proportion of each component of the TME was determined by applying the ESTIMATE algorithm to all LUAD samples, resulting in three scores: ImmuneScore, StromalScore, and ESTIMATEScore. The relationship between the three scores and the risk score was evaluated using the R packages “limma” and “ggpubr”, and a violin plot was created. Using the R packages “ggplot2”, “ggpubr”, and “ggExtra”, we examined the correlations between the risk scores and TME by using Spearman correlation analysis. Next, the penetration values of the LUAD samples were calculated based on seven algorithms: XCELL, TIMER, QUANTISEQ, MCPCOUNTER, EPIC, CIBERSORT-ABS, and CIBERSORT. We then performed Spearman correlation analysis utilizing the R packages “limma”, “scales”, “ggplot2”, “ggtext”, “reshape2”, “tidyverse”, and “ggpubr” to assess the relationship between immune cell subpopulations and risk score values and in order to draw bubble plots. Furthermore, ssGSEA was performed based on the CIBERSORT and preprocessCore algorithms to investigate the relationships between risk score, LUAD immune cell infiltration, and immune cells. We used the R packages “limma”, “GSEABase”, and “GSVA” to analyze the relationship between the infiltration of 16 immune cells and 13 immunity functions and the risk score. Finally, we also compared the activation of immune checkpoints (ICs) between different risk groups using the R packages “ggpubr” and “ggplot2” and the plotted box-line plots.

### 2.9. Relationships between Risk Scores, Immunotherapy, and Drug Sensitivity

The Tumor Immune Dysfunction and Exclusion algorithm (TIDE) (http://tide.dfci.harvard.edu) (accessed on 8 July 2022) could predict the response of different risk groups to ICI treatment. TCGA LUAD expression profiles were first normalized using the R package “scale”. Next, the TIDE online tool was used to calculate the TIDE score, Dysfunction score, and Exclusion score, and the R packages “limma” and “ggpubr” were used to evaluate their relationship with the risk scores and to draw violin plots. We assessed their associations with the risk scores using the R packages “limma” and “ggpubr”, and violin plots were produced. The Cancer Immunome Atlas (TCIA; https://tcia.at/home) (accessed on 12 July 2022) provided us with the TCGA LUAD expression matrix and ICI (anti-PD1 and anti-CTLA4) scores. The R packages “limma” and “ggpubr” were applied to estimate the relationship between the ICI treatment monitoring risk scores and the TCIA scores for anti-PD1 and anti-CTLA4 inhibitors, alone or in combination, in high- and low-risk groups. The R package “pRRophetic” was used to build a ridge regression model with tenfold cross-validation to infer semi-inhibitory concentration (IC50) values for the analysis of therapeutic biomarkers in the Genomics of Drug Sensitivity in Cancer database (GDSC; https://www.cancerrxgene.org/) (accessed on 15 July 2022) and in order to draw violin plots. The DNA and RNA stemness indices were calculated using a one-class logistic regression (OCLR) machine learning algorithm to examine the relationship between risk scores and stemness scores.

### 2.10. Relationship between Risk Score and GSVA

We used the R packages “limma”, “GSEABase”, “GSVA”, “reshape2”, and “ggplot2” to analyze the GSVA enrichment of DENRGs in the prognostic model as well as the Spearman correlation test to examine correlations between these genes and the enriched pathways.

### 2.11. DENRGs Associated with Early Diagnosis of LUAD Was Used to Construct a Diagnostic Model

Based on their clinical information, pathological stage 1, stage 2, and standard samples were screened for further analysis on the GSE75037 chip. We used the SangerBox software (SangerBox 1.0.9) to extract the expression matrices of target genes closely related to early diagnosis in the prognostic model. Then, a binary logistic regression analysis was performed using SPSS to construct an early diagnosis model, and the diagnostic efficacy of the model was analyzed with the use of ROC curves.

### 2.12. Online Database

To validate the correlation between nine DENRGs and clinicopathological staging, UALCAN (http://ualcan.path.uab.edu/analysis.html) (accessed on 25 June 2022) was analyzed. We used the GSCA (http://bioinfo.life.hust.edu.cn/GSCA/#/) (accessed on 28 June 2022) database to explore the relationship between prognostic model-related DENRGs and the drug IC50. The Human Protein Atlas database (HPA; http://www.proteinatlas.org/) (accessed on 20 June 2022) was used to validate the expression of nine DENRGs in LUAD tissues as compared with normal tissues. The TISIDB (http://cis.hku.hk/TISIDB/index.php) (accessed on 1 July 2022) was used to examine the expression of PANX1 in its presentation in different immune isoforms of LUAD.

## 3. Results

### 3.1. Identification of DENRGs and Enrichment Analysis Related to DENRGs

The differential expression analysis of 734 NRGs (526 LUAD tissues versus 327 normal tissues) revealed 208 DENRGs (99 upregulated and 109 downregulated genes) (Figure 2). The GO enrichment analysis revealed that DENRGs were primarily associated with intracytoplasmic translation, oxidative stress response, cellular scorching, and apoptotic signaling pathways (Figure 3A,C). The KEGG enrichment analysis of DENRGs focused on signaling pathways, including necroptosis, NOD-like receptors, apoptosis, tumor necrosis factor, IL-17, and NF-κB (Figure 3B,D).

### 3.2. Construction and Validation of the LUAD Prognostic Model

The whole TCGA LUAD dataset was randomly divided into training and test sets at the ratio of 7:3. First, a survival analysis was performed on the training set using the univariate Cox regression analysis and the K-M method for each DENRG, and genes with *p* < 0.05 were selected as candidate markers. Thirty-seven DENRGs associated with OS were screened in the training set (Figure 4A). Secondly, we used the Lasso regression analysis to screen 19 prognosis-related DENRGs out of 37 DENRGs to avoid overfitting the prognostic model (Figure 4B). Finally, a LUAD prognostic model consisting of nine DENRGs was constructed using multivariate Cox regression analysis, and the following risk score formula was also derived:

RiskScore = (0.53357 × TMEM44) + (0.38098 × ZNF146) + (0.48044 × FAF2) + (0.37444 × PANX1) + (0.33212 × MLKL) + (0.38542 × PPIA) + (0.43709 × PMAIP1) + (0.23428 × TRAF2) + (−0.30466 × PLCG1) (Figure 4C).

Based on whether the hazard ratio (HR) was greater than 1, we determined that PLCG1 was a protective factor (HR < 1), while the remaining eight DENRGs were risk factors (HR > 1). Each patient in the training set was divided into high- and low-risk groups based on the median risk score. In the training set, the K–M method was used to determine the effect of the risk score on the prognosis, and it was found that LUAD patients in the low-risk group had a longer OS than those in the high-risk group, with AUC values of 0.758, 0.745, and 0.826 at 1, 3, and 5 years, respectively (Figure 4D,E). The distribution of risk scores and survival status can be seen in Figure 4F. The PCA results of the training set show that the high-risk and low-risk groups were well-separated in both directions, indicating that the risk score had an excellent discriminatory ability (Figure 4G). Next, we performed three validations of the LUAD prognostic model constructed from the training set: the TCGA internal test set, the TCGA full set, and the GSE19188 dataset. First, according to the internal test validation, patients in the high-risk group had a poorer prognosis (*p* < 0.001), while the low-risk group had a better prognosis, with AUC values of 0.638, 0.704, and 0.772 at 1, 3, and 5 years, respectively (Figure 5A,B). The risk scores, survival status distribution, and PCA results are shown in Figure 5C,D. Second, in the TCGA full-set validation, the low-risk group of LUAD patients still had a longer OS than the high-risk group, with AUC values of 0.719, 0.732, and 0.797 at 1, 3, and 5 years, respectively (Figure 5E,F). Finally, in the GSE19188 dataset validation, the prognosis for the high-risk group remained worse (*p* = 0.0013), with AUC values of 0.6, 0.676, and 0.726 at 1, 3, and 5 years, respectively (Figure 5G,H). By comparison, we found that all three validation results remained highly consistent with the predictions of the training set. According to the results of this study, the LUAD prognostic model has a more stable predictive ability.

### 3.3. Independent Predictive Value of Risk Scores and Clinical Characteristics

Univariate and multivariate Cox regression analyses were performed to investigate whether risk scores and multiple clinical characteristics could be used as independent predictors of OS in patients with LUAD. In the training group, the univariate Cox analysis revealed that risk score (HR = 1.720, 95% confidence interval (CI) = 1.516–1.951, *p* < 0.001), tumor stage (HR = 1.685, 95% CI = 1.389–2.044, *p* < 0.001), T (HR = 1.541, 95% CI = 1.185–2.004, *p* = 0.001), and N (HR = 2.221, 95% CI = 1.718–2.871, *p* < 0.001) were all associated with OS in patients with LUAD (Figure 6A). Our multivariate Cox regression analysis showed that our risk score (HR = 1.622, 95% CI = 1.413–1.861, *p* < 0.001) was an independent prognostic indicator (Figure 6B). The receiver operating characteristic (ROC) and the C-index were used to assess the predictive power of risk scores and multiple clinical characteristics. It was found that the prognostic model had a predictive accuracy AUC of 0.777, which was better than that for other clinical characteristics (such as age (0.577), sex (0.577), and TMN (0.666, 0.498, and 0.758, respectively)) (Figure 6C,D). For the training set, we created a heat map of clinical features (Figure 6E). To provide clinicians with a more quantitative means of predicting the prognosis of LUAD, we combined the risk score with other clinical features to construct a clinically useful nomogram. According to the results, the risk score was the most significant factor among the clinical parameters (Figure 6F). In this study, most patients with LUAD had a total score of between 340 and 480, which was calculated by summing the scores of the individual indicators. Figure 6 illustrates the risk value for one patient as an example. Our calibration curve, ROC, and DCA validation results indicate that our nomogram accurately predicts total OS (Figure 6G–I).

### 3.4. Correlation of Risk Scores with GSEA and Mutations

According to the GSEA, the prognostic models were significantly enriched in regulatory immune, cell cycle, and tumor-related pathways, primarily the proteasome, the P53 pathway, steroid biosynthesis, the pentose phosphate pathway, arachidonic acid metabolism, and the DNA transcription, replication, and repair processes (Figure 7A,B). Then, we compared the risk scores of somatic mutation driver genes between the two groups by plotting the top 20 most frequently mutated DENRG using waterfall plots. The findings revealed that the same DENRG were mutated at different frequencies in different risk groups, and that these genes were mutated more frequently in the high-risk group (Figure 7C,D). Therefore, we believe the above results may provide a new direction for therapeutic studies with regard to gene mutations.

### 3.5. Relationship between Risk Scores and TME, Immune Cells, and ICs

NRGs are associated with immune cell infiltration, which is closely linked to tumor development and prognosis; hence, we explored the relationship between risk scores and the ratio of immune cells to stromal components. There was a significant difference between groups in terms of StromalScore, ImmuneScore, and ESTIMATEScore (Figure 8A). Interestingly, the three scores were negatively correlated with the risk score, indicating that low-risk patients had higher tumor immunoreactivity than high-risk patients (Figure 8B–D). By assessing the potential relationship between immune cell subsets and risk scores, it is possible to determine whether prognostic model-related DENRGs are involved in the tumor immune microenvironment (TIME). Immunoinfiltration analysis using seven algorithms, including TIMER, XCELL, QUANTISEQ, MCPCOUNTER, EPIC, Cibersort-ABS, and CIBERSORT, showed significant correlation between changes in immune cell landscape and different risk groups (Figure 8E). Following that, we assessed the enrichment fractions of 16 immune cell types and the activity of 13 immune-related functions in the low-risk and high-risk groups. The differences in aDCs, B_cells, DCs, iDCs, Mast_cells, Neutrophils, pDCs, T_helper_cells, and TIL between the two groups were significant (*p* < 0.05), and the rate of immune cell infiltration was higher in the low-risk group population (Figure 9A). In addition, for immune-related functions, HLA, MHC_class_I, and Type_II_IFN_Response were significant (*p* < 0.05), particularly in the low-risk group (Figure 9B). Programmed death receptors and their ligands are referred to as ICs. It is possible to inhibit the binding of programmed death receptors and their ligands using therapeutic approaches based on ICs for ICI. This can enhance the host immune system’s aggressiveness against tumor cells and inhibit tumor development and progression. The results show that some IC genes were differentially expressed in different risk groups. Among them, the expression levels of CD276, TNFSF9, and TNFRSF9 were higher in the high-risk group than in the low-risk group, suggesting that the poor prognosis of high-risk patients may be partly due to the immunosuppressive microenvironment (Figure 9C). The findings show a significant correlation between risk scores and TME, immune cells, and ICs in LUAD, an essential guideline for future immunotherapy studies.

### 3.6. Relationship of Risk Score with Immunotherapy and Drug Sensitivity

The TIDE algorithm was used to predict the effects of immunotherapy from transcriptomic data to determine whether prognostic models could predict the efficacy of immunotherapy in patients with LUAD. The higher the TIDE score, the greater the likelihood of immune escape and the less effective the ICI treatment. Our results showed TIDE scores that were lower in the low-risk group than those in the high-risk group, indicating that ICI treatment was more effective in the low-risk group (Figure 10A). Furthermore, there were significant differences in immune dysfunction scores (Dysfunction) and immune rejection scores (Exclusion) between the two groups (Figure 10B,C). Thus, the low-risk group may have tremendous potential for ICI treatment. TCIA is a comprehensive immunogenomic analysis based on TCGA that correlates risk scores with patients’ immunotherapy outcomes by immunophenotype scores (IPS). According to our findings, full IPS and CTLA4 inhibitor IPS were significantly lower in the high-risk group than in the low-risk group, particularly CTLA4 inhibitor IPS, which strongly predicted an inadequate response to immunotherapy in patients with higher risk scores. However, the PD-L1 inhibitor IPS and PD1+CTLA4 inhibitor IPS did not differ significantly between the two risk groups (Figure 10D). Next, the utility of risk scores for guiding treatment aspects was also explored. Our first step was to select drugs commonly used in first-line chemotherapy regimens for LUAD, including vincristine, cisplatin, paclitaxel, docetaxel, and gemcitabine. Then, to assess the response to chemotherapy in these two risk groups, drugs widely used in LUAD chemotherapy, including etoposide, cisplatin, erlotinib, vincristine, methotrexate, and bleomycin, were added. The high-risk group was more sensitive to seven anti-cancer drugs (paclitaxel, docetaxel, cisplatin, gemcitabine, etoposide, vincristine, and vincristine), and the IC50 of most chemotherapeutic drugs was significantly lower in the high-risk group than that in the low-risk group, suggesting that the use of these drugs may be more effective in high-risk patients (Figure 10E). The expression of stemness-related biomarkers in tumor cells is highly correlated with drug resistance, cancer recurrence, and tumor proliferation. Therefore, we evaluated the correlation between the DNA stemness score (DNAss), the RNA stemness score (RNAss), and the risk score. Both the DNAss and the RNAss were positively correlated with risk scores, indicating that the group with higher risk scores had a greater stemness capacity, and that the correlation was stronger for RNAss (R = 0.49, *p* < 0.001) than for DNAss (R = 0.093, *p* = 0.05) (Figure 10F,G). Overall, this risk score was more effective in predicting chemotherapy drug sensitivity in high-risk patients, whereas low-risk patients were better treated with immunotherapy.

### 3.7. Comprehensive Analysis of Nine DENRGs in the Prognostic Model

Since the risk score of LUAD showed a strong correlation with various aspects, such as immunotherapy, drug sensitivity, and TME, we further explored each gene’s potential role in the risk score through a comprehensive analysis of multiple aspects. Initially, we compared the differential expressions of nine DENRGs in LUAD with the histochemical results (Figure 11). Except for MLKL, the immunohistochemical results for the remaining eight DENRGs were contained in the HPA database (protein expression of MLKL could not be retrieved from HPA data) (see Appendix A). As predicted by the prognostic model, all DENRGs were deleterious factors (HR > 1), except for PLCG1, which was protective (HR < 1). We then explored the correlation of nine DENRGs with immune cells and found that TRAF2, PPIA, and PMAIP1 were all negatively associated with immune cells, whereas PANX1 and MLKL mostly had a positive association (Figure 12A). TMB and MSI are predictive markers of cancer immunotherapy efficacy, and to clarify the critical role of the nine DENRGs in LUAD, we explored their correlation with TMB and MSI in LUAD. The results show that PANX1 (*p* < 0.001), PMAIP1 (*p* = 0.002), PPIA (*p* < 0.001), TRAF2 (*p* < 0.001), and ZNF146 (*p* < 0.001) were all associated with TMB (Figure 12B), while MLKL (*p* = 0.001), PLCG1 (*p* < 0.001), TMEM44 (*p* < 0.001), TRAF2 (*p* < 0.001), and ZNF146 (*p* = 0.001) were all associated with MSI (Figure 12C). The expression of all DENRGs in LUAD increased with increasing TMB scores and MSI scores. Finally, the GSCA database was used to analyze the expression of nine DENRGs and the drug IC50 for drug sensitivity analysis, and the top 30 drugs were plotted. The results show that low levels of PMAIP1, PPIA, and ZNF146 were associated with resistance to most chemotherapeutic agents in GDSC and CTRP, whereas high levels of MLKL, TMEM44, FAF2, and PANX1 were associated with drug sensitivity (Figure 13A,B). Our findings suggest that these nine prognosis-related DENRGs could serve as biomarkers for drug screening and as new therapeutic targets for clinical treatment development.

### 3.8. Building LUAD Diagnostic Models

In exploring whether the expression of nine DENRGs in the prognostic model is significant at different pathological stages in LUAD patients, we found that the expression of all these genes was significantly associated with the clinicopathological stages 1 and 2 of LUAD (Figure 14A) (see Appendix A). Since stages I and II are typically considered early stages of tumors, exploring the presence of dual biomarkers that are both diagnostic and prognostic in prognostic models opens up new avenues for the clinical studies of LUAD. Using the GSE75037 microarray as the training set, we extracted the expression matrix for nine DENRGs from stage 1, 2, and the normal groups. A binary logistic regression was conducted on the training set to obtain a diagnostic model for LUAD, which consisted of four DENRGs (MLKL, PANX1, TRAF2, and PMAIP1).

Y(RiskScore) = −32.825 + (−4.948 × ∆CtMLKL) + (3.4888 × ∆CtPANX1) + (4.750 × ∆CtTRAF2) + (1.348 × ∆CtPMAIP1), with a combined ROC of 99.4% (Figure 14B).

In the training set, MLKL, PANX1, PMAIP1, and TRAF2 had AUC values of 0.139, 0.781, 0.743, and 0.92, respectively. According to the TCGA LUAD test set validation, the combined ROC reached 99.0%, and the AUC values for MLKL, PANX1, PMAIP1, and TRAF2 were 0.080, 0.904, 0.957, and 0.929, respectively (Figure 14C). Finally, we concluded that the diagnostic model could be used to achieve an excellent early diagnosis of LUAD and an excellent early independent diagnosis of PANX1 and TRAF2 (see Table 2 for details).

### 3.9. Hub Genes with Both Diagnosis and Prognosis

According to the GEPIA database, all the genes, except for TRAF2, showed independent survival significance in the prognostic model (Figure 15A). GSVA conducted a further investigation using a non-parametric unsupervised analysis method to determine whether genomic enrichment was associated with different pathways. The results show that PANX1 was significantly and positively correlated with most pathways, most prominently with the P53 pathway (Figure 15B). We were also surprised that risk scores were primarily enriched in the P53 pathway, suggesting that PANX1 may be a representative gene in the prognostic model. Since TRAF2 (*p* = 0.26) had no independent survival significance and no independent prognostic power in the multifactorial Cox regression (*p* = 0.137), we concluded that TRAF2 could be used as a separate diagnosis indicator. At this point, we also found that PANX1 expression was closely correlated with a prognostic risk score, and that PANX1 expression was stronger in the high-risk group, which is consistent with the previous analysis (Figure 16A). By combining the above multiple analyses, this study was able to establish that PANX1 had both independent survival significance (*p* = 0.017) and excellent independent diagnostic and prognostic ability (*p* = 0.023), and that the high expression of PANX1 was closely associated with high-risk populations. Thus, it is reasonable to conclude that PANX1 can serve as an independent diagnostic and prognostic biomarker for LUAD and warrants further investigation.

### 3.10. Immunocorrelation Analysis of PANX1

We used the TISIDB website to investigate the relationship between PANX1 expression and LUAD immune subtypes. There are six immune subtypes, including C1 (wound healing), C2 (IFN-gamma-dominant), C3 (inflammatory), C4 (lymphocyte-depleted), C5 (immunologically quiet), and C6 (TGF-b-dominant). According to the results, PANX1 expression was associated with the immune subtype of LUAD, with a low expression in C3 and a high expression in C6 (Figure 16B). When the relationship between PANX1 and TME was investigated, a significant difference in StromalScore and ESTIMATEScore was found between the groups (Figure 16C). PANX1 was positively correlated with T cell CD4 memory activated, Macrophages M1, Neutrophils, and T cell CD4 memory resting, and it was negatively associated with Mast cell resting and T cells follicular helper (Figure 16D,E). Among the frequently tested ICs, PANX1 had a relatively strong positive correlation with CD276, PDCD1LG2, PD-L1 (CD274), TNFSF4, and TNFRSF9 (Figure 16F), and it had a negative correlation with TNFRSF14. Finally, the immune efficacy results show that PANX1 was more effective in low-risk patients treated with the CTLA4 inhibitor IPS alone than in high-risk patients (Figure 16G). These findings suggest that PANX1 is closely associated with immunity and may be a novel target for immunotherapy in LUAD, with a positive predictive power for future clinical diagnosis, prognosis, and immunotherapy.

## 4. Discussion

In this study, nine DENRGs were used to construct a prognostic model for LUAD. Patients with LUAD were categorized into high-risk and low-risk groups using the median value of their risk scores. Patients in the low-risk group had a significantly better prognosis than those in the high-risk group, as demonstrated by the results of both the training and test groups, with AUC values of 0.758, 0.745, and 0.826, respectively, for 1-, 3-, and 5-year OS. The prognostic risk score was an independent predictor of OS in patients with LUAD, with a better predictive value for survival than other conventional clinical characteristics. Since no prognostic model of LUAD consisting of NRGs from other investigators has been identified, we compared it with a model composed of miRNAs that could indirectly regulate NRGs. The AUCs for the 3-year and 5-year OS in the LUAD prognostic model of seven miRNAs were 0.631 and 0.605, respectively [14]. Another study showed that the AUCs of the LUAD prognostic model composed of seven necroptosis-related lncRNAs were 0.723, 0.679, and 0.715 for 1-year, 3-year, and 5-year OS, respectively [15]. In addition, we compared this study to a non-NRG-constructed LUAD prognostic model, and the results from the LUAD prognostic model consisting of six mRNAs based on single-cell RNA-sequencing and bulk RNA-sequencing data show that the AUCs for 1-year, 3-year, and 5-year OS were 0.669, 0.674, and 0.642, respectively [16]. Through the above comparison, we found that the LUAD prognosis model constructed by NRGs in this study has a good predictive effect. Therefore, we believe that this model will provide a new reference for prognostic risk stratification assessment and treatment strategy selection for LUAD patients.

The molecular mechanisms of necroptosis have been elucidated in many degenerative and inflammatory diseases. For example, excess reactive oxygen species are produced during necroptosis, affecting β-amyloid production in Alzheimer’s disease [17,18]. Since tumor formation and immune escape are also closely related to the immune microenvironment, the role of NRGs in tumor formation, especially its role in immune regulation, is gaining widespread attention [19]. Because necroptosis regulates tumor immunity, we used ssGSEA to investigate the immune status of different risk populations. Some immune cells (aDCs, B_cells, DCs, iDCs, Mast_cells, Neutrophils, pDCs, T_helper_cells, and TIL) and immune functions (HLA, MHC_class_I, and Type_II_IFN_Reponse) were more active in the low-risk group, and some of them were closely associated with necroptosis. Evidence suggests that necrotrophic apoptotic cells provide tumor-specific antigens and inflammatory cytokines to DCs for antigen cross-triggering, which activates cytotoxic CD8+ T lymphocytes. RIPK3 is required to regulate cytokine expression in DCs and is potentially involved in innate and adaptive immunity [20]. Serine proteases are involved in the neutrophils’ RIPK3-MLKL-mediated necrotizing death pathway [21]. Based on our findings and on those of previous research, we further confirmed that necroptosis might be involved in the development of LUAD by regulating tumor immunity.

Immunotherapy has been a revolution in cancer treatment, improving the situation of patients with unresectable stages of disease [22]. TMB, PD-L1, PD-L2, and MSI are effective biomarkers for predicting the efficacy of immunotherapy. However, the relationship between these biomarkers is complex, and it is unclear whether combining them is superior to using a single marker [23,24]. Researchers have repeatedly reported that the expression levels of ICs are highly correlated with immunotherapy efficacy, which paves the way for future research into LUAD immunotherapy [25,26,27]. CTLA4 inhibitors have been shown to benefit patients with NSCLC in clinical trials [28]. Therefore, we examined the expression of some common ICs in relation to prognostic risk scores. In our study, patients in the low-risk group had higher levels of IC expression than those in the high-risk group. For example, CTLA4 expression was significantly higher in the low-risk group than in the high-risk group. We believe that ICI treatment may be more effective in low-risk patients. Previous studies have demonstrated that anti-CTLA4 and/or anti-PD-1 ICI therapies can treat clinically advanced tumors with promising results [29,30,31]. Therefore, we used the TIDE algorithm to assess the potential efficacy of ICI therapy in high-risk and low-risk populations. The higher the TIDE score, the greater the likelihood of immune escape and the lower the effectiveness of the ICI therapy. In this study, we found that the low-risk group with a low TIDE score may benefit more from ICI therapy than the high-risk group with a high TIDE score. In addition, there were significant differences in the immune rejection (Exclusion) and immune dysfunction (Dysfunction) scores between the risk groups, confirming the superior efficacy of ICI therapy in the low-risk group. Since ICI treatment information was unavailable in the TCGA LUAD dataset, IPS was used as a surrogate for ICI efficacy. IPS was designed to predict patient responses to anti-PD-1 and anti-CTLA4 treatments, and it was developed primarily using TCGA RNA-seq data [32]. As with previous findings, the low-risk group had higher overall IPS and IPS for CTLA4 inhibitors, confirming that low-risk populations respond more effectively to immunotherapy. In clinical practice, chemotherapy is another effective strategy for treating LUAD based on the predicted response to chemotherapy [33]. We found that almost all the first-line chemotherapeutic agents (vincristine, cisplatin, paclitaxel, docetaxel, and gemcitabine) performed better in patients with higher risk scores, and most had significantly lower IC50 values than they did in the low-risk group. As a result, we believe that prognostic risk scores are related to the response to chemotherapy and are more effective, particularly in high-risk patients. In conclusion, in terms of the treatment strategies for LUAD, it is reasonable to assume that patients in the high-risk group are more suitable for chemotherapy regimens, whereas patients in the low-risk group benefit more from immunotherapy.

P53 is a classical oncogene that is crucial for maintaining genomic integrity [34]. P53 regulates many biological processes such as cell cycle arrest, apoptosis, senescence, and metabolism. There is growing evidence that P53 also regulates innate and adaptive immune responses. P53 affects the innate immune system by secreting factors that regulate macrophage function to suppress tumorigenesis. When P53 is dysfunctional in cancer, it affects the recruitment and activity of T cells and myeloid cells, resulting in immune evasion [35]. In this study, the GSVA results revealed that the prognostic model was most closely linked to P53 signaling, with PANX1 contributing the most to this model. Meanwhile, the GSEA results showed that the prognostic model was equally enriched in the P53 pathway. Therefore, we suggest that the P53 pathway is closely linked to the NRGs pathway, and that the cause of the immune escape of cancer cells may be due to the joint regulation of the two pathways. However, the upstream and downstream relationships of the related genes remain to be studied in depth. The primary gene in the prognostic model, PANX1, was significantly correlated with drug sensitivity, TMB, and immune cell infiltration. In addition, PANX1 was also positively correlated with many frequently tested ICs. Recent studies have suggested that PANX1 may have a wide range of biological effects on cancer development, including the promotion of cell proliferation and tumorigenesis in melanoma, brain tumors, and hepatocellular carcinoma [36,37,38,39]. It has been reported that PANX1, ABC, CALHM1, VRACs, and MACs can regulate TME via ATP release channel modulation in order to exert therapeutic effects against cancer [40,41]. However, no results for PANX1 have been found in LUAD. In the model we constructed, PANX1 had better efficacy when treated with the CTLA4 inhibitor alone in low-risk patients, a result that is consistent with previous studies. Lastly, we created an early diagnosis model using the DENRGs associated with early diagnosis in the LUAD prognostic model. Additionally, this diagnostic model was highly effective in validating the TCGA LUAD dataset and the GSE19188 chip. At this point, we were surprised to find that PANX1 was a gene with highly independent diagnostic efficacy among the four mRNAs in the diagnostic model. In summary, we suggest that PANX1 is an independent risk factor in the early diagnosis and prognosis of LUAD, and that it is closely associated with immunotherapy and drug sensitivity. This novel finding leads us to believe that PANX1 deserves further exploration as a potential new LUAD target.

Despite the comprehensive data analysis and multiple data validations performed in our study, there are some limitations and shortcomings. First, it is well-known that identifying differentially expressed genes is very important to the final performance, which is similar to feature selection. In this study, we used Cox regression and Lasso to screen and combine differentially expressed genes, which proved to have a good predictive effect. With the continuous development of machine learning, increasing numbers of advanced technologies are being used to identify and screen differentially expressed genes. According to recent research, Modified Gray Wolf Optimizer (MGWO), Cross-view Local Structure Preserved Diversity and Consensus Learning (CVLP-DCL), Unsupervised Linear Feature Selective Projection (FSP), and other methods show excellent feature selection capabilities [42,43,44]. Therefore, these methods will be suitable for identifying and combining differentially expressed genes. In the future, we will continue combining clinical data and machine learning methods to explore more advanced prediction methods for clinical diagnosis, prognosis assessment, and treatment. In addition, all data were obtained from public databases, and the number of patients was limited. Therefore, this prognostic model needs to be validated with more clinical data. Finally, the complex mechanism of PANX1 in LUAD development remains unknown and requires further in-depth exploration using in vivo or in vitro experiments.

## 5. Conclusions

In this paper, a prognostic model for LUAD was constructed with the use of bioinformatics analysis. This model regulated the immune microenvironment, cell cycle, and DNA damage repair mechanisms. Risk scores were significantly correlated with ICI treatment and chemotherapeutic drug sensitivity. In addition, we identified a core gene, PANX1, that is useful in immune regulation, prognostic assessment, and early diagnosis. Finally, we believe that this study will provide new immunotherapy targets for LUAD and a new theoretical foundation for the clinical diagnosis, prognostic assessment, and individualized treatment of patients with LUAD.

## Figures and Tables

**Figure 1 cancers-14-05153-f001:**
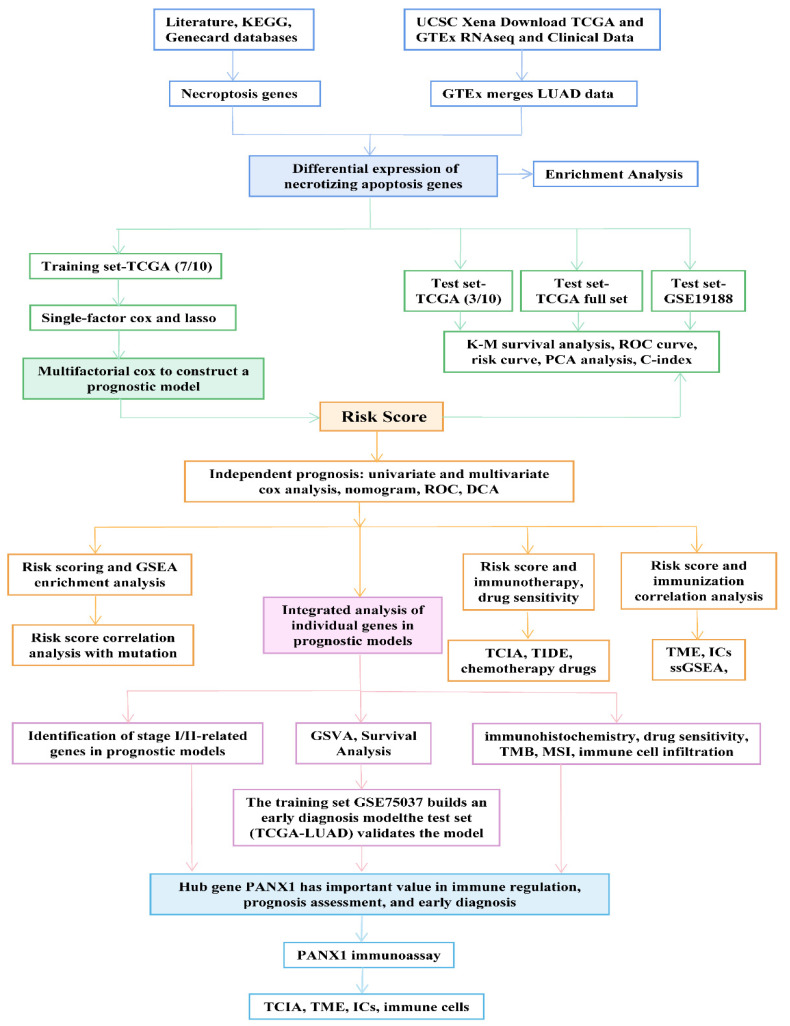
Overview of the analytical process of this study.

**Figure 2 cancers-14-05153-f002:**
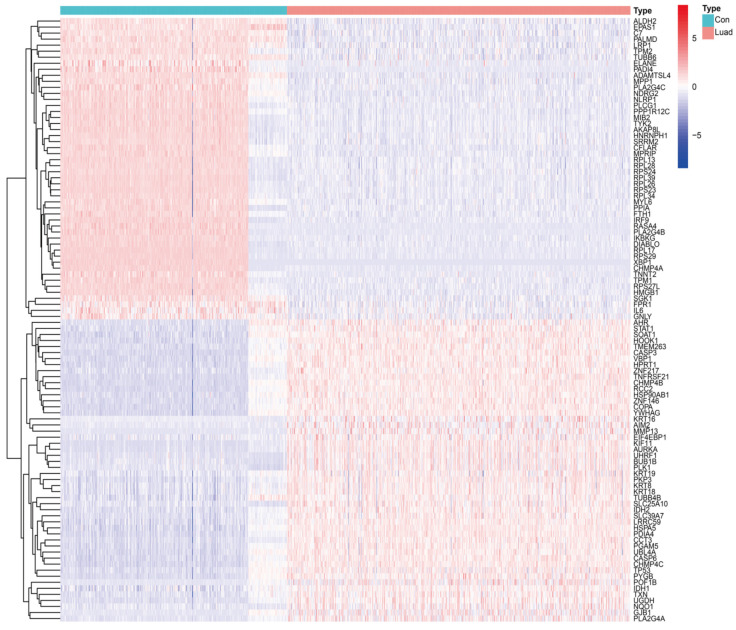
Heat map of NRG expression in LUAD and normal tissues.

**Figure 3 cancers-14-05153-f003:**
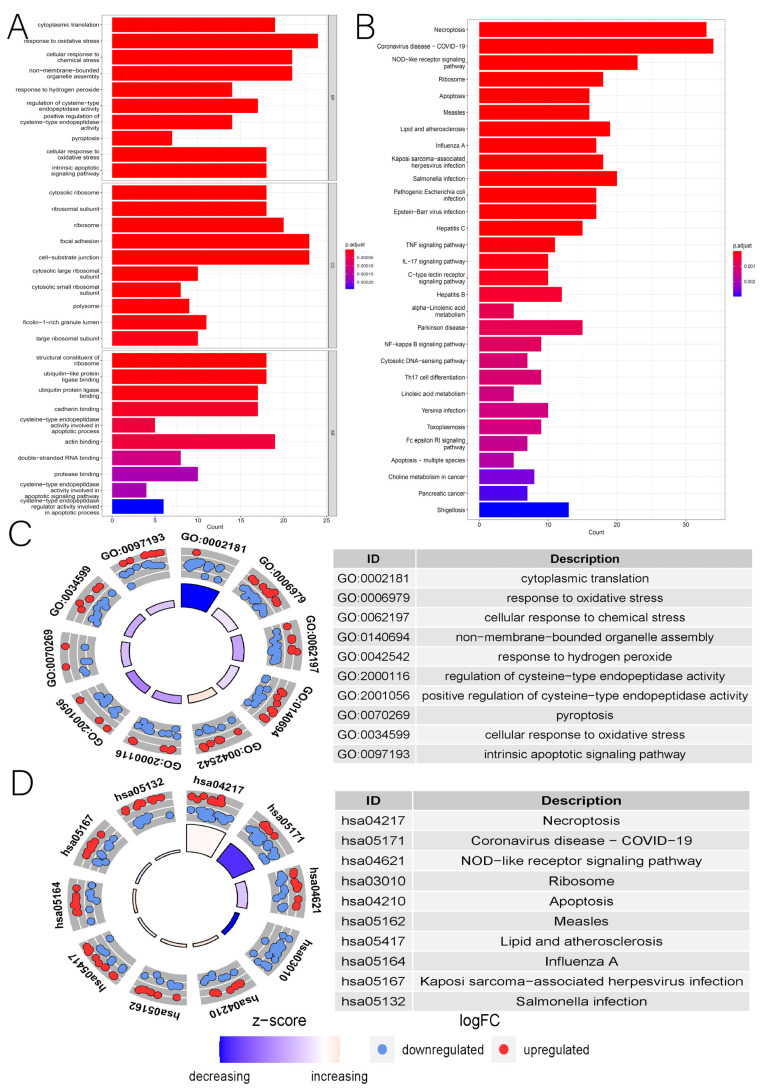
(**A**) Bar graph of GO enrichment analysis. (**B**) Bar graph of KEGG pathway enrichment analysis. (**C**) Circle diagram of GO enrichment analysis. (**D**) Circle diagram of KEGG pathway enrichment analysis.

**Figure 4 cancers-14-05153-f004:**
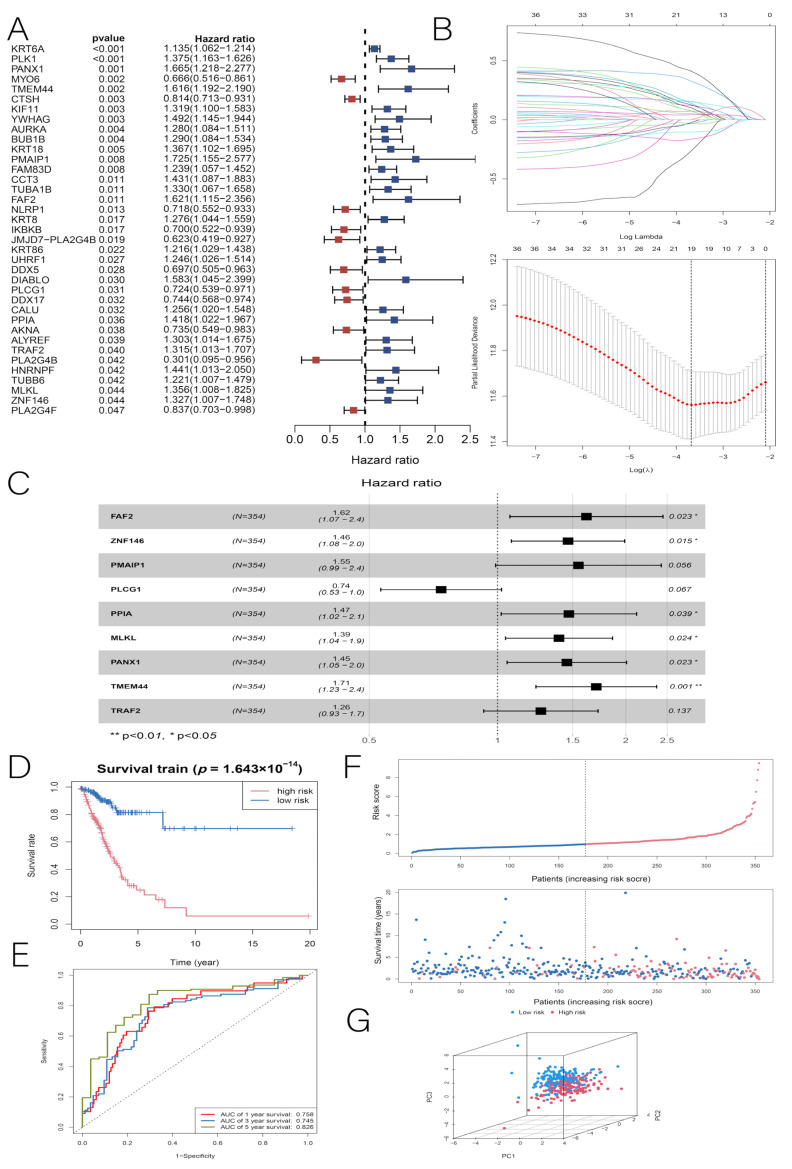
(**A**) Univariate Cox regression forest plot for the training group. (**B**) Inclusion of DENRGs in Lasso regression analysis to screen out prognosis-related DENRGs. (**C**) Multi-factor Cox regression forest plot for the training group. (**D**) K-M plots of OS for patients in the high-risk and low-risk groups in the training group. (**E**) ROC curves for the training group with 1-year, 3-year, and 5-year OS. (**F**) Risk survival status plots for the training group. (**G**) PCA distribution plot for the training group. * *p* < 0.05, ** *p* < 0.01.

**Figure 5 cancers-14-05153-f005:**
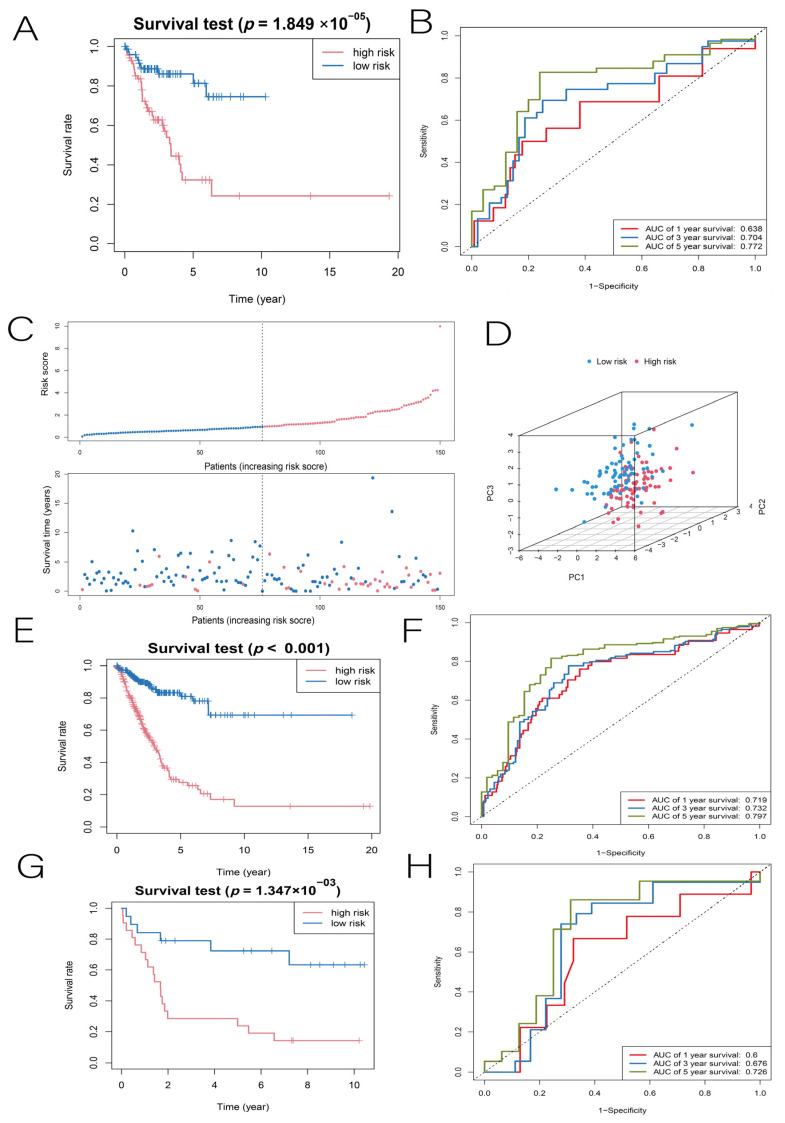
(**A**) In the internal test set, K-M plots of OS for patients in the high-risk and low-risk groups. (**B**) ROC curves for the internal test group with 1-year, 3-year, and 5-year OS. (**C**,**D**) Risk survival status maps and PCA distribution maps for the internal test group. (**E**,**F**) K-M plots of OS for patients in the high-risk and low-risk groups when the complete TCGA set was performed as a test set, and ROC curves for 1-year, 3-year, and 5-year OS. (**G**,**H**) K-M plots of OS for patients in the high-risk and low-risk groups when the GSE19188 chip was used for the test set, and ROC curves of OS at 1, 3, and 5 years.

**Figure 6 cancers-14-05153-f006:**
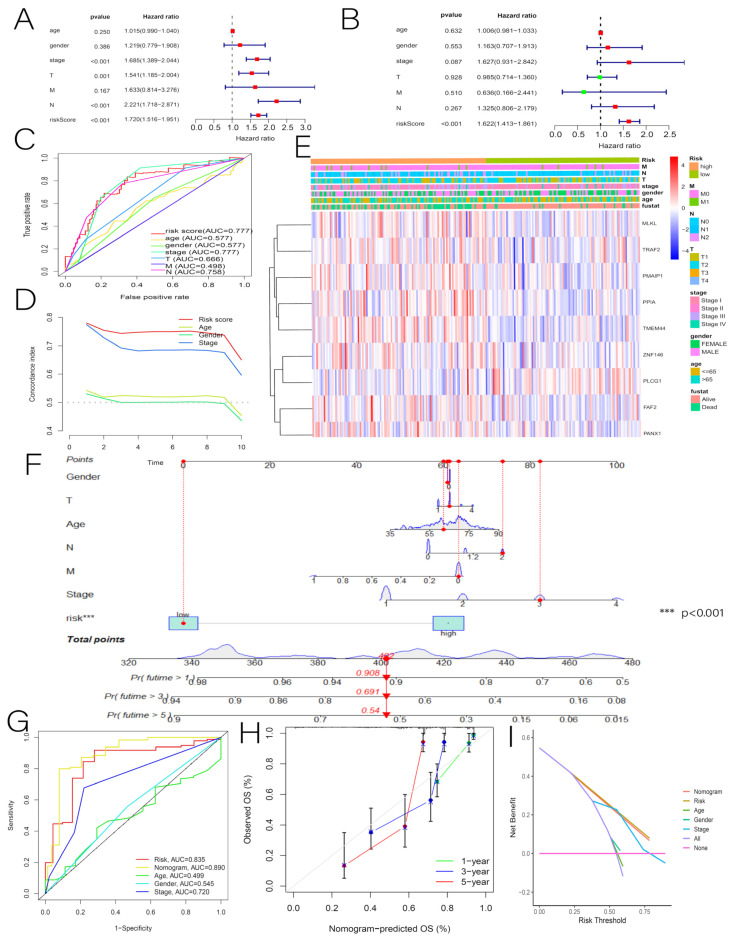
(**A**) Univariate Cox analysis of risk scores and multiple clinicopathological features. (**B**) Multivariate Cox analysis of risk scores and numerous clinicopathological features. (**C**,**D**) ROC and C-index were used to determine the independent predictive value of risk scores and clinicopathologic features. (**E**) Heat map of clinicopathological characteristics. (**F**) Construction of the nomogram for 1-year, 3-year, and 5-year OS. (**G**–**I**) Calibration curves, ROC, and DCA were used to determine the accuracy of the nomogram. *** *p* < 0.001.

**Figure 7 cancers-14-05153-f007:**
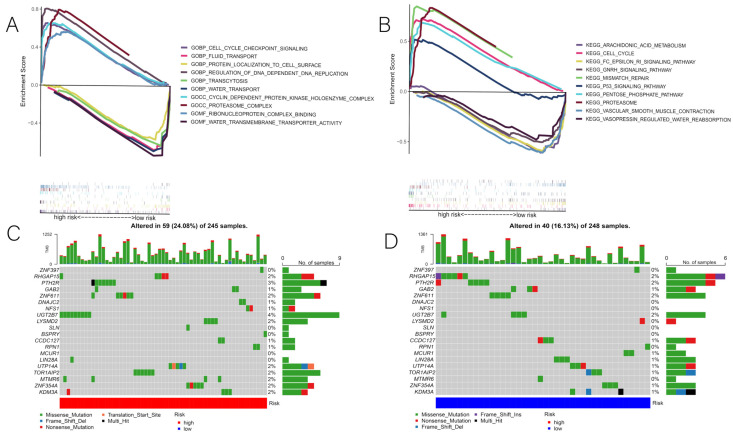
(**A**,**B**) Results of the GSEA analysis. (**C**,**D**) The top 20 driver genes with the highest mutation frequencies in the low-risk and high-risk groups.

**Figure 8 cancers-14-05153-f008:**
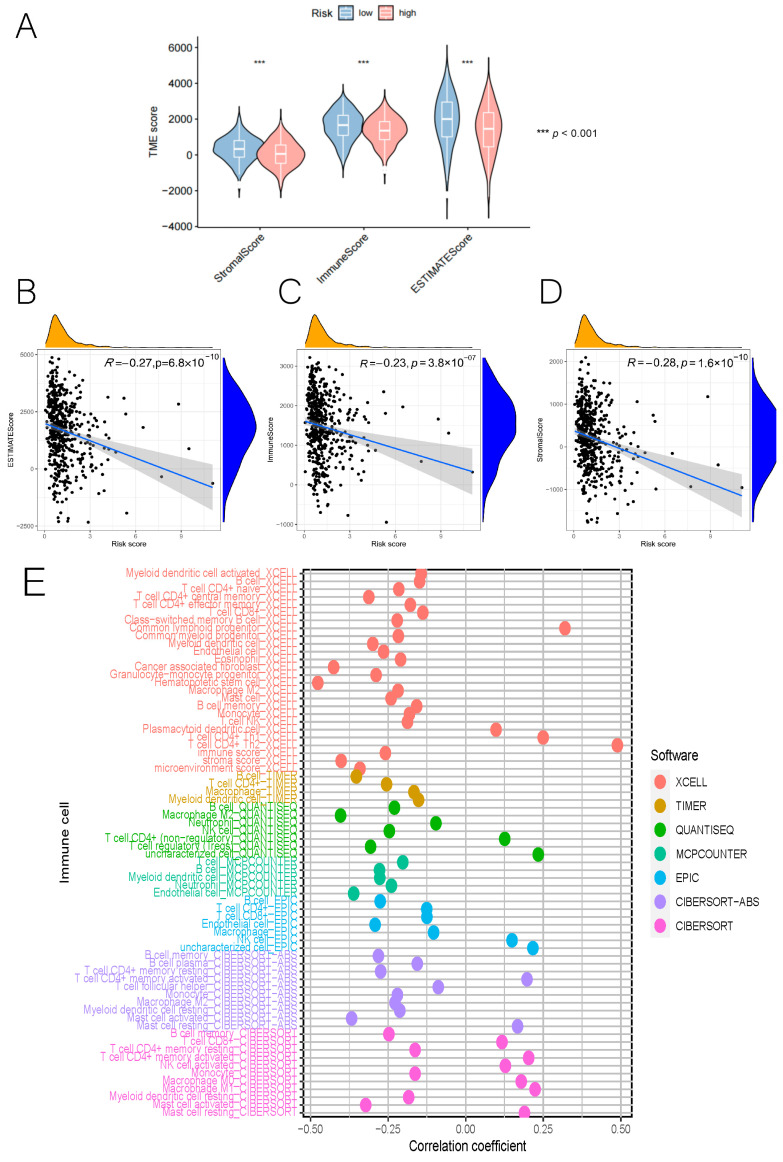
(**A**) Relationship between risk score and tumor microenvironment. (**B**–**D**) Correlation of StromalScore, ImmuneScore, and ESTIMATEScore with risk score. (**E**) Analysis of risk score correlation with immune cell infiltration based on 7 algorithms. *** *p* < 0.001.

**Figure 9 cancers-14-05153-f009:**
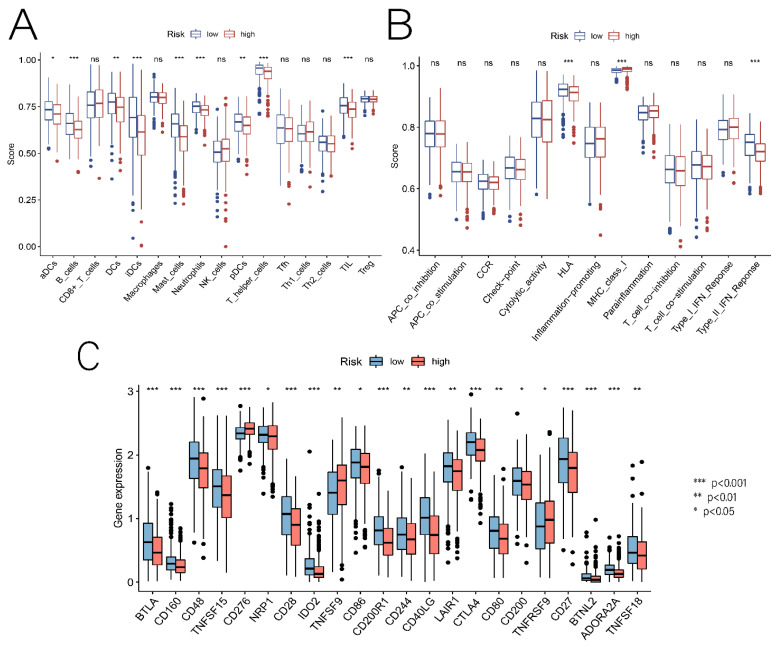
(**A**) Relationship between risk score and immune cell enrichment. (**B**) Relationship between risk score and immune cell-related function. (**C**) Relationship between risk scores and immune checkpoints. * *p* < 0.05, ** *p* < 0.01, and *** *p* < 0.001.

**Figure 10 cancers-14-05153-f010:**
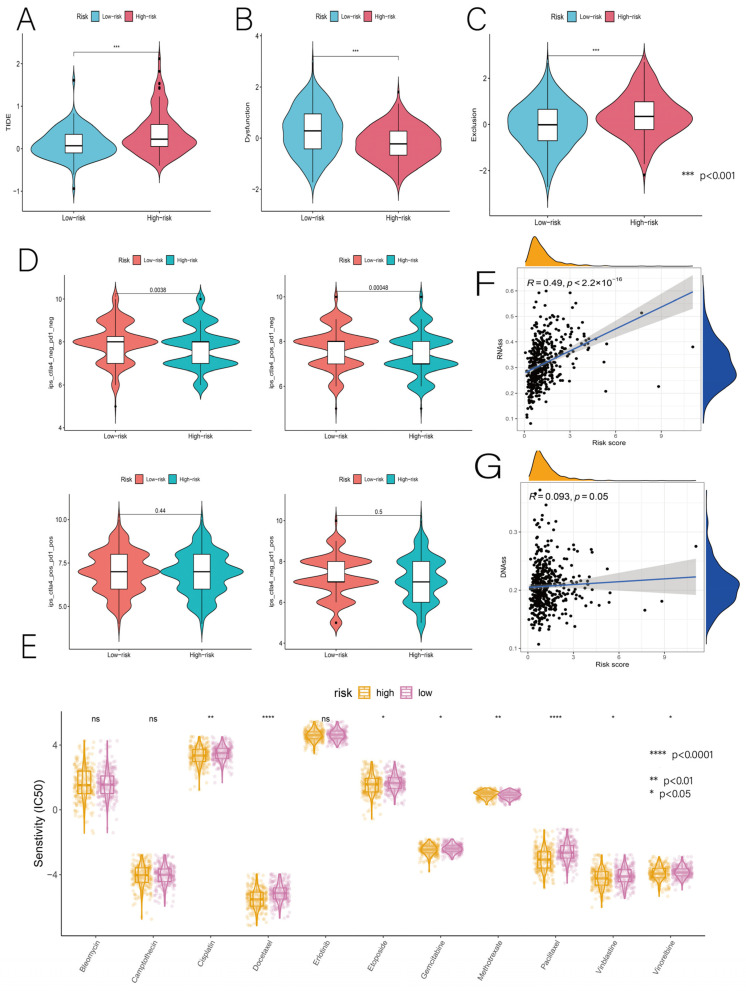
(**A**–**C**) Relationship between high-risk and low-risk groups and TIDE scores (including TIDE, Dysfunction, and Exclusion). (**D**) Differences in IPS between different risk groups. IPS and IPS-CTLA4 significantly differed between the high-risk and low-risk groups (*p* < 0.05). (**E**) Relationship between high-risk and low-risk groups with multiple chemotherapeutic agents commonly used for LUAD. (**F**,**G**) Relationship between high-risk and low-risk groups with DNAss and RNAss. * *p* < 0.05, ** *p* < 0.01, *** *p* < 0.001, and **** *p* < 0.0001.

**Figure 11 cancers-14-05153-f011:**
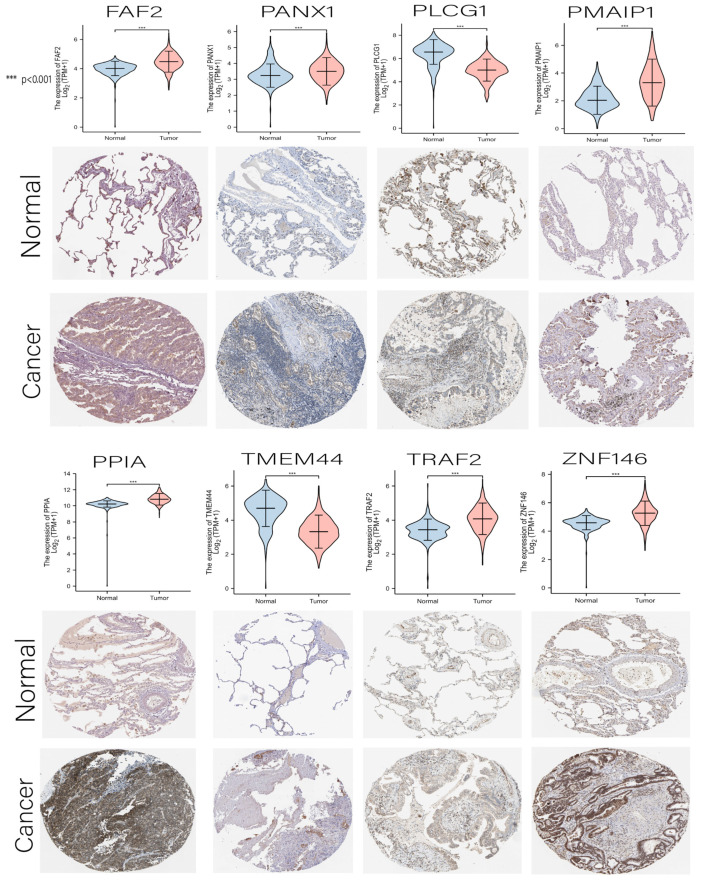
Differential expression map and immunohistochemical results of nine DENRGs in the prognostic model (MLKL is not available in the HPA database). *** *p* < 0.001.

**Figure 12 cancers-14-05153-f012:**
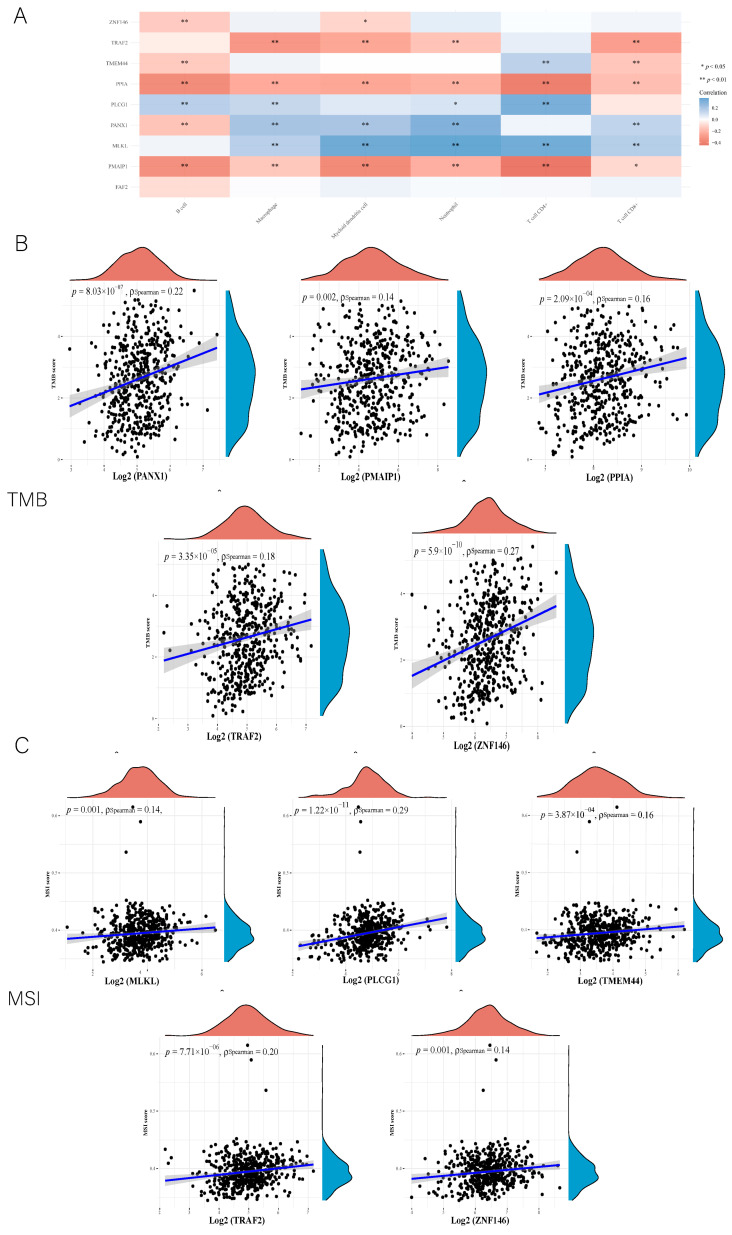
(**A**) Correlation of DENRGs with immune cells in the prognostic model. (**B**) DENRGs associated with TMB in the prognostic model. (**C**) DENRGs associated with MSI in the prognostic model. * *p* < 0.05, ** *p* < 0.01.

**Figure 13 cancers-14-05153-f013:**
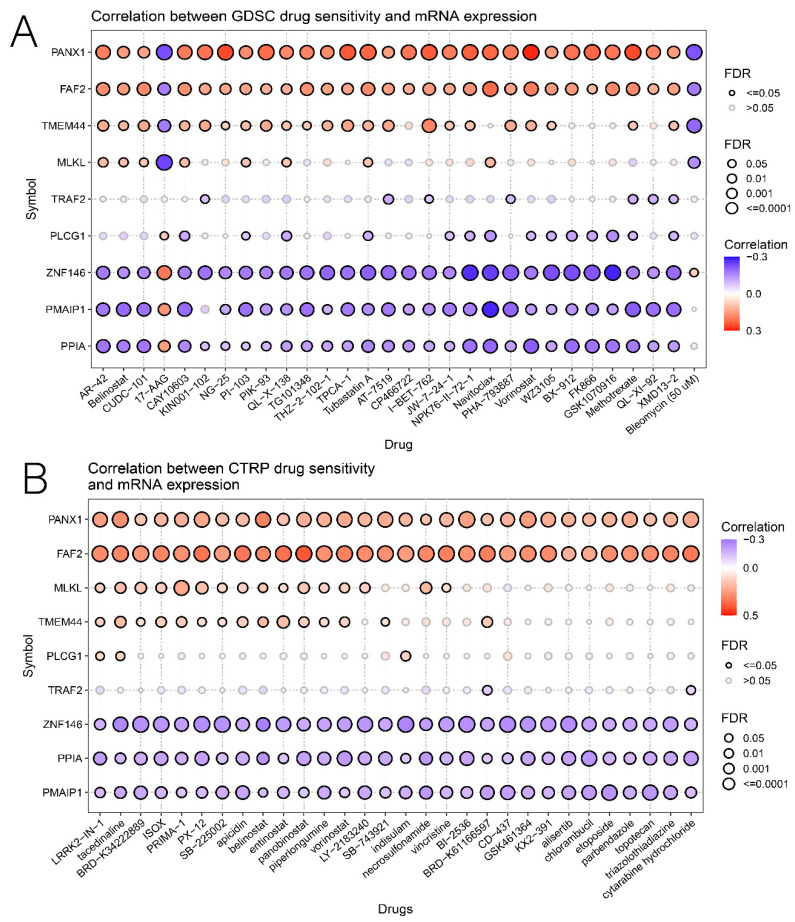
(**A**,**B**) Relationship between nine DENRGs in the prognostic model and multiple chemotherapeutic drugs in two drug sensitivity databases, CTRP and GDSC.

**Figure 14 cancers-14-05153-f014:**
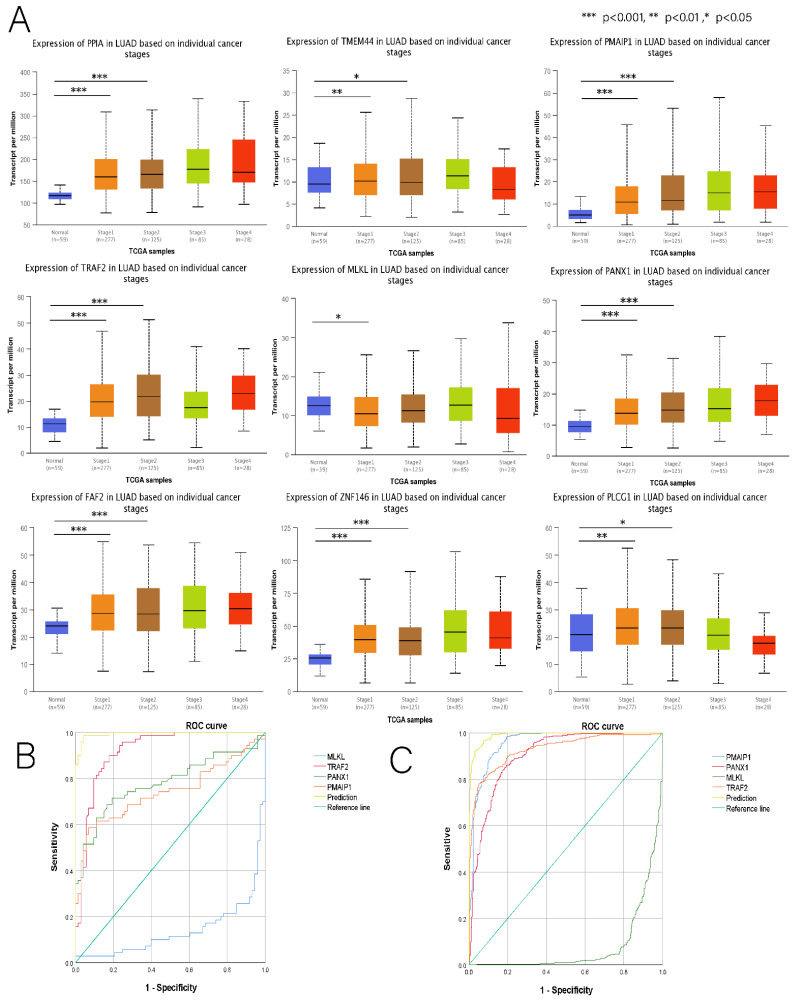
(**A**) Relationship between nine DENRGs and clinicopathological staging in the prognostic model. (**B**) ROC curves of the LUAD diagnostic model when GSE75037 was used for the training set. (**C**) ROC curves of LUAD when TCGA was used for the test set. * *p* < 0.05, ** *p* < 0.01, and *** *p* < 0.001.

**Figure 15 cancers-14-05153-f015:**
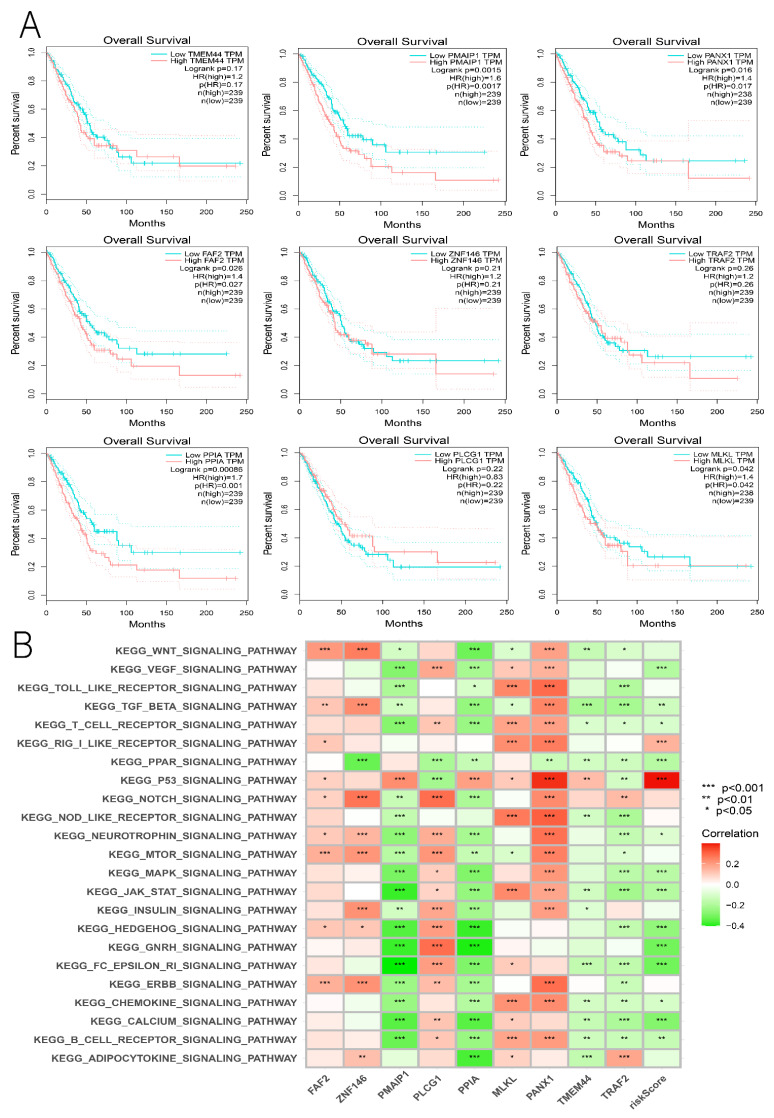
(**A**) Survival analysis of nine DENRGs in the LUAD prognostic model. (**B**) GSVA analysis of nine DENRGs and prognostic risk scores. * *p* < 0.05, ** *p* < 0.01, and *** *p* < 0.001.

**Figure 16 cancers-14-05153-f016:**
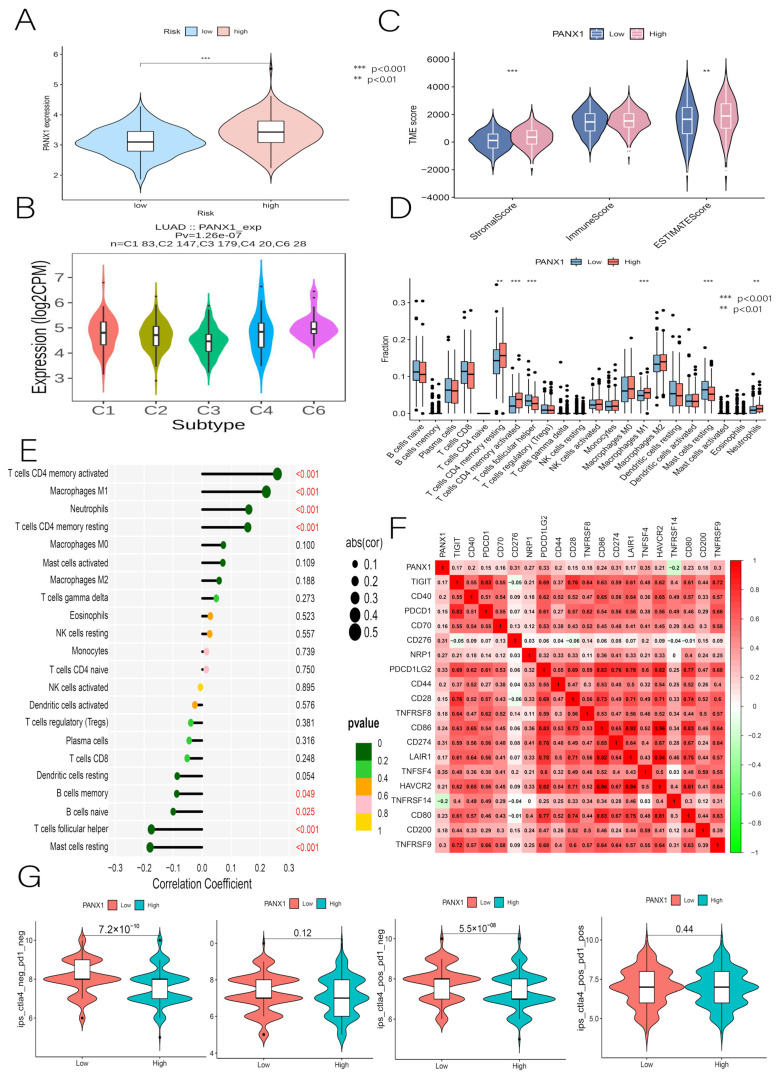
(**A**) Correlation of PANX1 with prognostic risk score. (**B**) Immunophenotyping of PANX1. (**C**) Relationship between PANX1 and TME. (**D**,**E**) Relationship between PANX1 and immune cell infiltration. (**F**) PANX1 and immune checkpoints. (**G**) Relationship between PANX1 and TCIA immunotherapy. ** *p* < 0.01, *** *p* < 0.001.

**Table 1 cancers-14-05153-t001:** Clinical information.

		TCGA-LUAD	GEO-GSE75037	GEO-GSE19188
Variable	Category	Numbers		
**Gender**	**Male**	352	48	66
	**Female**	242	118	22
**Diagnostic Age**	**≤65**	265	58	Unknown
	**>65**	310	108	Unknown
**Stage**	**I**	174	50	40
	**II**	324	20	Unknown
	**III**	71	11	Unknown
	**IV**	23	2	Unknown
**T**	**T1**	23	Unknown	Unknown
	**T2**	93	Unknown	Unknown
	**T3**	200	Unknown	Unknown
	**T4**	119	Unknown	Unknown
**M**	**M0**	445	Unknown	Unknown
	**M1**	16	Unknown	Unknown
**N**	**N0**	383	Unknown	Unknown
	**N1**	127	Unknown	Unknown
	**N2**	67	Unknown	Unknown
	**N3**	5	Unknown	Unknown
**Fustat**	**Alive**	354	83	86
	**Dead**	240	83	24

**Table 2 cancers-14-05153-t002:** Accuracy of early diagnosis.

**Train-GSE75037**	**Normal**	**LUAD**	**All**	**Accuracy**
**4-mRNA negative**	70	3	73	95.9
**4-mRNA positive**	2	68	70	97.1
**All accuracy**	70	68	143	96.5
**Test-TCGALUAD**	**Normal**	**LUAD**	**All**	**Accuracy**
**4-mRNA negative**	321	26	347	92.5
**4-mRNA positive**	14	368	382	96.3
**All accuracy**	321	368	729	94.5

## Data Availability

The data presented in this study are available on request from the corresponding author.

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
