# Peer review of "Construction of a Prognostic and Early Diagnosis Model for LUAD Based on Necroptosis Gene Signature and Exploration of Immunotherapy Potential"

_cancers, 2022, doi:10.3390/cancers14205153_

Round 1
Reviewer 1 Report
This paper proposes a prognostic model to predict prognostic outcomes for patients with LUAD, which can regulate the immune microenvironment, the cell cycle, and DNA damage repair mechanisms. Extensive experiments verify the effectiveness of the proposed method. However, there are still some problems need to be addressed, and my main concerns are listed as follows.
1.The authors should refine the simple summary and abstract, and the current version of them are too long.
2.This paper contains many abbreviations of professional terms, the authors are suggested to use a table to put these abbreviations’ introduction at the front of the paper, not at the end.
3.There are many figures in the paper, while the size of some figures are not appropriate. The authors should adjust them to make the contents of paper more compact, such as Figure 2D, 2E and Figure 7.
4.The motivations and contributions of this paper are suggested to be clearly stated. For example, the authors can list them in points, which makes the organization of paper more fluent.
5.In table 2, what is the meaning of all accuracy? The all accuracy of train-GSE75037 is the average value of 4-MRNA negative and positive, why the all accuracy of test-TCGALUAD is not?
6.It is well known that Identification of differentially expressed genes is important for final performance, which is very similar to feature selection. Can the commonly used feature selection methods be used for this work, following are some related works that should be cited for discussion: Cross-view Locality Preserved Diversity and Consensus Learning for Multi-view Unsupervised Feature Selection, TKDE 2021;Gene selection for microarray data classification based on Gray Wolf Optimizer enhanced with TRIZ-inspired operators, KBS 2021.Feature Selective Projection with Low-Rank Embedding and Dual Laplacian Regularization, TKDE 2020;
7.In figure 11, the texts of all subfigures are completely unreadable, and the authors should enlarge their resolution.
Author Response
Response to Reviewer 1 Comments
Dear Reviewer,
Thank you for your comment on the Construction of a Prognostic and Early Diagnosis Model for LUAD Based on Necroptosis Gene Signature and Exploration of Immunotherapy Potential of Commenting on the article (cancers-1924240). These comments are of great value and help us to revise and improve our articles. In response to your comments, we have carefully studied the comments and made corrections. We hope to get your approval. The main revisions and replies to reviewers are as follows:
Point 1: The authors should refine the simple summary and abstract, and the current version of them are too long.
Response 1: We quite agree with your suggestion. The content of simple summary and abstract is really too long. In view of this suggestion, we have summarized and refined the content of simple summary and abstract.
Point 2: This paper contains many abbreviations of professional terms, the authors are suggested to use a table to put these abbreviations’ introduction at the front of the paper, not at the end.
Response 2: We attach great importance to your suggestion. Putting abbreviations at the end of the article makes readers have reading difficulties, so I sorted out the abbreviations in the whole text and put them at the back of the abstract. But when resubmitting the revised draft, I have a lot of reading the magazine's nearly 30 large data analysis class articles, and I found that all article's abbreviations are placed at the end. I think it may be associated with the magazine layout form, so I put the abbreviations back at the end, but that doesn't mean I don't want to. I still very adopt the proposal. I want to make further adjustments in the following revision according to the requirements of you, the editor, and the magazine.
Point 3: There are many figures in the paper, while the size of some figures are not appropriate. The authors should adjust them to make the contents of paper more compact, such as Figure 2D, 2E and Figure 7.
Response 3: Thank you for your advice. Many of the images in this article are indeed dyslexic. Because the image is too small or too low resolution problem, it becomes unclear after inserting Word. In view of this suggestion, we have modified the original 2D, 2E and Figure 7 according to the requirements of the magazine, and updated the picture numbers to 2C, 2D and Figure 8.
Point 4: The motivations and contributions of this paper are suggested to be clearly stated. For example, the authors can list them in points, which makes the organization of paper more fluent.
Response 4: We appreciate your suggestion very much. The motivation and contribution of this paper are not clear enough. For this suggestion, we have explained the motivation of this article in the first and second paragraphs of the introduction. In addition, we have divided the main contributions of this paper into three points and summarized them in a simple summary, which can facilitate readers to understand them more intuitively and quickly. Moreover, we have made detailed explanations of these three main contributions in the result section (3.2, 3.8, 3.9, 3.10).
Point 5: In table 2, what is the meaning of all accuracy? The all accuracy of train-GSE75037 is the average value of 4-MRNA negative and positive, why the all accuracy of test-TCGALUAD is not?
Response 5: Thank you for your suggestion. This part is indeed our negligence. The data in the third row of Table 2 was omitted in the previously submitted manuscript, and we have filled it in again. All accuracy represents the percentage of the total number of people with the sum of all correctly predicted people, rather than averaging the results of 4-mRNA negatives and positives. Therefore, in Train-GSE75037, the calculation formula of all accuracy is (70+68) /143, and the result is 96.5%. In Test-TCGA LUAD, the calculation formula of all accuracy is (321+368) /729, and the result is 94.5%.
|
Train -GSE75037 |
Normal |
LUAD |
All |
Accuracy |
|
4-mRNA negative |
70 |
3 |
73 |
95.9 |
|
4-mRNA positive |
2 |
68 |
70 |
97.1 |
|
All accuracy |
70 |
68 |
143 |
96.5 |
|
|
||||
|
Test -TCGALUAD |
Normal |
LUAD |
All |
Accuracy |
|
4-mRNA negative |
321 |
26 |
347 |
92.5 |
|
4-mRNA positive |
14 |
368 |
382 |
96.3 |
|
All accuracy |
321 |
368 |
729 |
94.5 |
Point 6: It is well known that Identification of differentially expressed genes is important for final performance, which is very similar to feature selection. Can the commonly used feature selection methods be used for this work, following are some related works that should be cited for discussion: Cross-view Locality Preserved Diversity and Consensus Learning for Multi-view Unsupervised Feature Selection, TKDE 2021;Gene selection for microarray data classification based on Gray Wolf Optimizer enhanced with TRIZ-inspired operators, KBS 2021.Feature Selective Projection with Low-Rank Embedding and Dual Laplacian Regularization, TKDE 2020;
Response 6: Thank you very much for your suggestion. We have read the three articles you recommended and learned more feature selection methods, which we did not know before, which is very inspiring. The feature selection method also applies to identifying differentially expressed genes and combinations. We have added to this in the last paragraph of our discussion (limitations and drawbacks). In scientific research work of the future, we will use clinical data combined with these advanced methods, further building the prediction method of clinical more meaningful and effective
Point 7: In figure 11, the texts of all subfigures are completely unreadable, and the authors should enlarge their resolution.
Response 7: Thanks for your suggestion. Our negligence is that the text of the subpicture in Picture 11 is unreadable. In view of this deficiency, we have adjusted and replaced the clarity of the picture, and the number of the new picture has been changed to Picture 14.
In addition to the above comments, we have adopted the English editing service of MDPI to correct the grammar and other problems in this manuscript.

Reviewer 2 Report
Strength of the study
The manuscript provides significant evidence for new immunotherapy targets for the LUAD. In this study, authors have established a prognostic model of LUAD based on necroptosis-related genes. This model tends to regulate the immune microenvironment, cell cycle, and DNA damage repair mechanism. The PANX1 gene was identified as a core gene that can be useful in immune regulation, prognostic assessment, and early diagnosis. The evidence provided in this manuscript is self-explanatory. The manuscript can be accepted for publication.
Limitations of the study
Most of the figures are hard to read in terms of their labeling, numbers, font size, values, and p values. This makes it the reader difficult to interpret the result and understand the resulting outcome. All figures need better representation.
Please see the suggestions below
Table 1- The table should be better formed to see the numbers side by side for understanding. My suggestion makes the variable bold or in italic so it can easily distinguish from its category variable or the table can be modified in a better representative way.
Here is an example
|
|
|
TCGA-LUAD |
GEO -GSE75037 |
GEO-GSE19188 |
|
Variable |
Category |
Numbers |
||
|
Gender |
Male |
352 |
48 |
66 |
|
Female |
242 |
118 |
22 |
|
|
Diagnostic Age |
≤65 |
… |
… |
… |
|
≥65 |
… |
… |
… |
|
|
Stage |
I |
|
|
|
|
II |
|
|
|
|
|
III |
|
|
|
|
|
IV |
|
|
|
|
For grade variables, all numbers are unknown in TCGA-LUAD, GEO-GSE75037, and GEO-GSE19188. It would be nice to omit the Grade from this table and explain this in the material and method section. Please explain the age factor and its relationship with LUAD with suitable references in the material and method section.
Figure 2A - Heatmap genes are unreadable. It will be more suitable to address the type as a Con and LUCD instead of Con and Treat to better relate with the manuscript. Please make the gene’s name in readable font for this heatmap.
Figure 2B-2E - The GO and KEGG pathways included in the bar as well as the circle diagram are not clear. Please improve the font, size, and clarity of all words for all the labels and pathways in the figure. It’s hard to visualize the figure labels.
Figure 2 legend- B and C are bar diagrams and D and E are circle diagrams for GO and KEGG pathways. It’s switched in the Figure 2 legend. Should be corrected.
Figure 3B and F - The X and Y axis font size for both sections are small and unreadable. Should be improved for the visualization purpose of this figure.
Figures 4B, C, F, and H - Unreadable font and should be clear.
Figure 5 – Should be clear as suggested above for all figures.
Figure 6 – Needs improvement in terms of visualization.
Figure 7 – Needs improvement as suggested above. Figure 7G the X axis should be shown properly. Please arrange this.
Figures 8, 9, 10, 11, and 12- Please make them clear and better versions figures.
Author Response
Dear Reviewer,
Thank you for your comment on the Construction of a Prognostic and Early Diagnosis Model for LUAD Based on Necroptosis Gene Signature and Exploration of Immunotherapy Potential of Commenting on the article (cancers-1924240). These comments are of great value and help us to revise and improve our articles. In response to your comments, we have carefully studied the comments and made corrections. We hope to get your approval. The main revisions and responses to reviewers are in the annexes.

Reviewer 3 Report
The research article entitled “Construction of LUAD prognostic and early diagnosis model based on necroptosis gene signature and exploration of immunotherapy potential” investigates of the necroptosis gene signature for early diagnosis of lung adenocarcinoma and Immunotherapy Potential. The article has many grammatical and sentence errors, and the language organization needs to be improved. For these reasons, I conclude that the paper should undergo minor revision.
1. Authors need to improve the abstract with key findings like gene expressing date with quantitative data
2. The introduction is good but very general in nature. More elaborately.
3. Revise the Introduction and carefully review the existing and recent Literature.
https://doi.org/10.1038/s41598-022-20217-4
https://doi.org/10.1038/s41598-022-15854-8
4. All the figure resolution needs to be improved
5. Typographical errors can be avoided. The language and grammar used throughout the manuscript need to be improved. Specific attention needs to be given to this which will improve the standard of the manuscript.
Author Response
Response to Reviewer 2 Comments
Dear Reviewer,
Thank you for your comment on the Construction of a Prognostic and Early Diagnosis Model for LUAD Based on Necroptosis Gene Signature and Exploration of Immunotherapy Potential of Commenting on the article (cancers-1924240). These comments are of great value and help us to revise and improve our articles. In response to your comments, we have carefully studied the comments and made corrections. We hope to get your approval. The main revisions and replies to reviewers are as follows:
Point 1: Authors need to improve the abstract with key findings like gene expressing date with quantitative data.
Response 1: We quite agree with your suggestion. In the abstract, some details are indeed lacking. Due to the limitation of no more than 250 words, we refined and modified the abstract repeatedly and presented some key findings intuitively in the form of data.
Point 2: The introduction is good but very general in nature. More elaborately.
Response 2: Thank you very much for your affirmation. Our introduction has shortcomings, and our motivation, contribution, and current research status are not well presented in the article. In response to this suggestion, we have changed the content of the introduction section to the current three-paragraph format. The first paragraph explains the clinical status of LUAD and our research motivation. The second paragraph mainly explains the research status of necroptosis, the significance of the research, and the possible influence of necroptosis on LUAD. The third paragraph explains the main research line of this paper and summarizes the main contributions.
Point 3: Revise the Introduction and carefully review the existing and recent Literature.
https://doi.org/10.1038/s41598-022-20217-4
https://doi.org/10.1038/s41598-022-15854-8
Response 3: Thank you very much for your advice. We have read the two articles you repeatedly recommended, especially the content of the introduction part. We replaced the previous two-paragraph introduction with a three-paragraph introduction to clarify the content structure. At the same time, by referring to these two literatures, we add some content to improve the introduction part of this paper. Thank you very much for your guidance and help.
Point 4: All the figure resolution needs to be improved.
Response 4: Thank you for your suggestion. It is our negligence that all the pictures are not clear. The previous image was too small or low-resolution, making it very unclear when inserted in Word. Given this deficiency, we did our best to adjust and replace the sharpness of all the pictures according to the magazine's requirements.
Point 5: Typographical errors can be avoided. The language and grammar used throughout the manuscript need to be improved. Specific attention needs to be given to this which will improve the standard of the manuscript.
Response 5: Thank you very much for your suggestion. In view of this deficiency, we have adopted the English editing service of MDPI to modify the grammar and other problems of this manuscript.
